# Classic Psychedelics and Human–Animal Relations

**DOI:** 10.3390/ijerph19138114

**Published:** 2022-07-01

**Authors:** Elin Pöllänen, Walter Osika, Cecilia U. D. Stenfors, Otto Simonsson

**Affiliations:** 1Centre for Psychiatry Research, Department of Clinical Neuroscience, Karolinska Institutet, 171 76 Stockholm, Sweden; walter.osika@ki.se (W.O.); otto.simonsson@ki.se (O.S.); 2Center for Social Sustainability, Department of Neurobiology, Care Sciences & Society, Karolinska Institutet, 141 83 Huddinge, Sweden; 3Department of Psychology, Stockholm University, 114 19 Stockholm, Sweden; cecilia.stenfors@psychology.su.se; 4Department of Sociology, University of Oxford, Oxford OX1 1JD, UK

**Keywords:** nature, human–animal relations, psychedelics, psilocybin, LSD

## Abstract

Previous research has found associations between classic psychedelic use and nature-relatedness, but the link between classic psychedelic use and human–animal relations remains largely unexplored. Using data representative of the US adult population, with regard to age, sex and ethnicity (N = 2822), this pre-registered study assessed lifetime classic psychedelic use, ego dissolution during respondents’ most intense experience using a classic psychedelic, and three measures related to human–animal relations: speciesism, animal solidarity and desire to help animals. The results showed that lifetime classic psychedelic use was negatively associated with speciesism (*β* = −0.07, *p* = 0.002), and positively associated with animal solidarity (*β* = 0.04, *p* = 0.041), but no association was found with desire to help animals (*β* = 0.01, *p* = 0.542). Ego dissolution during the respondents’ most intense experience using a classic psychedelic was negatively associated with speciesism (*β* = −0.17, *p* < 0.001), and positively associated with animal solidarity (*β* = 0.18, *p* < 0.001) and desire to help animals (*β* = 0.10, *p* = 0.007). The findings indicate that classic psychedelics and ego dissolution may have an impact on human–animal relations. As these results cannot demonstrate causality, however, future studies should use longitudinal research designs to further explore the potential causal link between classic psychedelic use and human–animal relations.

## 1. Introduction

In a relatively short amount of time, human activities have changed the biosphere of our planet. The biomass of humans and livestock (mainly cattle and pigs) far surpasses that of wild mammals [1], a current reality that is driving climate change, environmental degradation and biodiversity loss [2,3,4]. This development can largely be traced to historical events and processes such as the domestication of livestock and the rise of industrialized agricultural practices, but the present rate and scale of exploitation of natural resources and animals, and its impact on earth, is unprecedented [1]. 

An increasing amount of sustainability research highlights the importance for humanity of reconnecting with nature and other living beings [5]. According to the biophilia hypothesis, humans have an innate need to connect with the natural world [6], and it has been argued that nature connectedness is a basic human psychological need [7]. A sense of connectedness to the natural world can lead to a unifying experience with nature [8], a “commitment to protect the self” [9], and pro-environmental attitudes and behaviors [10,11]. Interaction with other animals can serve as a bridge to re-connect with nature [12]. Animals and humans can develop strong bonds and caring for an individual animal, and conditions affecting the animal’s wellbeing can expand environmental concerns [13]. Seeing from another animal’s perspective, and thereby decreasing the boundaries between the self and the animal, has also been shown to increase pro-environmental concern [14]. 

### 1.1. Human–Animal Relations 

While humans biologically belong to the animal group, human identity is commonly characterized by a cultural separation from animals, largely based on human exceptionalism and prejudice toward animals in favor of the perceived interests of members of one’s own species, also referred to as speciesism [15,16]. Human attitudes toward—and treatment of—animals differ according to the animal species, which shapes our everyday interaction with animals [17,18], as well as conservation and welfare efforts, where species perceived to be similar to humans (e.g., sharing bio-behavioral traits with humans) are favored [19]. The framework of human–animal relations, where some animals are valued (e.g., companion animals) and others are de-individualized/de-valued (e.g., farm animals) [20,21,22], may give rise to cognitive dissonance (when values and behaviors conflict), which can trigger defense mechanisms [23]. In those instances, one can change values, behavior, or how the behavior is perceived so it aligns with values [24,25,26]. For instance, dissonance is reduced and disassociation is increased by denying (food) animals’ intellectual and emotional capacities [3,27]. 

Previous studies have found that animal solidarity in humans, characterized by a psychological bond with and commitment to animals as in-group members, predicts more pro-social behaviors and attitudes toward different types of animals, and a stronger opposition to animal exploitation [28]. People with greater solidarity with animals have been shown to have a greater desire to help animals in a more altruistic and empowering way, even in cases where it might imply fewer human resources and privileges. Greater solidarity with animals has also predicted higher ambivalence to meat eating [21] and less prejudice toward animals (i.e., speciesism), as well as towards humans (e.g., sexism, ageism, racism) [29]. Conversely, a greater perceived hierarchical human–animal divide has been linked to out-group prejudice and dehumanizing tendencies [30,31]. In experimental research, being presented with highly processed meat has been found to increase disassociation, whilst food dishes, pictures, or language use reminding people of the animal-meat connection (e.g., by displaying the whole animal body; saying cow instead of beef) has been found to increase empathy for the animal, and increase disgust toward consuming such meat [24]. 

Previous attempts to address sustainability issues, at the magnitude and rate needed to reach international and national established sustainability goals, have failed [32,33]. Effective interventions to alter human–animal relations could play a key role in changing behaviors, as well as developing frameworks that address the underlying drivers of environmental problems, with efforts that manage to avoid disastrous and long-term social and environmental disruption [9,21,34,35,36,37]. It is therefore important to investigate novel interventions with the potential to impact human–animal relations. 

### 1.2. Classic Psychedelics and Nature Relatedness

In recent years, psychedelic serotonin 2A receptor agonists (“classic psychedelics”), combined with psychological support, have been explored as a novel treatment model for a range of psychiatric disorders [38,39]. Although definitions vary slightly, classic psychedelics typically include psilocybin (“magic mushrooms”), *N*,*N*-dimethyltryptamine (DMT), the DMT-containing concoction ayahuasca, lysergic acid diethylamide (LSD), mescaline, and the mescaline-containing cacti peyote and San Pedro [40]. The evidence to date suggests that the potential mental health benefits of classic psychedelic use may be mediated by the quality of the acute experience, which has been measured by constructs such as ego dissolution [41,42,43]. 

Ego dissolution refers to a loss of self-identity, or to a dissolving of the boundaries between oneself and one’s surroundings. Such a phenomenon (i.e., ego dissolution) is not uncommon during the acute classic psychedelic experience [44]. The degree of self-reported ego dissolution during a classic psychedelic experience has been linked to alterations in brain connectivity [42,45], and has also been associated with wellbeing-related outcomes [41,44]. 

While research on classic psychedelics has primarily focused on mental health-related issues, recent research suggests that classic psychedelic use may have effects on environmental attitudes and behaviors. For example, one open-label pilot study found that psilocybin-assisted therapy produced sustained increases in nature-relatedness [46]. Another study also found increases in nature-relatedness following classic psychedelic use, with effects dependent on the degree of ego dissolution during the acute classic psychedelic experience [43]. Such findings correspond with results from cross-sectional research on classic psychedelic use and nature-relatedness [47] (see also [48]). 

### 1.3. The Present Study

Despite research on the link between classic psychedelic use and nature-relatedness, the relationship between classic psychedelic use and human–animal relations remains largely unexplored. Using data representative of the US adult population, with regard to age, sex and ethnicity (N = 2822), this pre-registered study assessed classic psychedelic use, ego dissolution during the respondents’ most intense experience using a classic psychedelic, and three measures related to human–animal relations: speciesism, animal solidarity and desire to help animals. We hypothesized that lifetime classic psychedelic use would be negatively associated with speciesist attitudes and positively associated with solidarity with animals and desire to help animals. We also hypothesized that ego dissolution during the respondents’ most intense experience using a classic psychedelic would be negatively associated with speciesist attitudes and positively associated with solidarity with animals and desire to help animals. 

## 2. Materials and Methods

### 2.1. Participants

The design plan, sampling plan, hypothesis and variables for this study were all pre-registered on the Open Science Framework: https://osf.io/4vtqk (pre-registered on 1 October 2021). The sample size was determined using Gpower (linear multiple regression, fixed model, R2 increase). 395 lifetime classic psychedelic users would achieve 80% power to detect small effect sizes with an alpha of 0.05. Assuming similar prevalence of lifetime classic psychedelic use in the US adult population as prior research (~14%; see [49]), we estimated that 2800 total participants would be necessary to get approximately 395 lifetime classic psychedelic users in the sample. The aim was therefore to recruit approximately 2800 participants. The sample (US residents, 18 years or older) was recruited through Prolific Academic (https://app.prolific.co) between the 1st and 9th of October 2021. Prolific Academic offers a representativeness function that uses proportionate stratification on three census-matched factors—sex, age and ethnicity—to reflect the demographic distribution of the US adult population.

Some questions, including demographic characteristics and human–animal relations, were asked to all respondents, whereas only respondents who reported lifetime classic psychedelic use (*n* = 613) were asked additional questions regarding ego dissolution during their most intense classic psychedelic experience. If the respondent completed the survey, they received a $2.20 payment. Study procedures were determined to be exempt from review by the Institutional Review Board at the University of Wisconsin-Madison. 

### 2.2. Variables

#### 2.2.1. Dependent Variables 

Speciesism [50] was assessed with five items on a 1–7 Likert-type scale (1 = strongly disagree, 7 = strongly agree); “*We should always elevate human interests over the interests of animals*”, “*When human interests conflict with animal interests, human interests should always be given priority*”, “*We should strive to alleviate human suffering before alleviating the suffering of animals*”, “*The suffering of animals is just as important as the suffering of humans*” (Reverse scored), and “*Having extended basic rights to minorities and women, it is now time to extend them also to animals*” (Reverse scored). Higher scores indicated more speciesist attitudes and endorsement, prioritizing human interests over animal interests. Internal consistency was good (*α* = 0.87). 

Solidary with animals [28] was assessed with five items on a 1–7 Likert-type scale (1 = strongly disagree, 7 = strongly agree); “*I feel a strong bond toward other animals*”; “*I feel solidarity toward animals*”; *and* “*I feel committed toward animals*”, “*I feel close to other animals*” and “*I feel a strong connection to other animals*”. Higher scores indicated more animal solidarity. Internal consistency was excellent (*α* = 0.96). 

Desire to help animals [21] was assessed with four items on a 1–7 Likert-type scale (1 = strongly disagree, 7 = strongly agree); “*I think it is very important to help animals*”, “*The government should adopt more policies and regulations to protect animals*”, “*Threatened species need to be preserved*”, and “*Caring about animals (pets, farm animals) is a part of my lifestyle.*” Higher scores indicate more desire to help animals. Internal consistency was acceptable (*α* = 0.77).

#### 2.2.2. Independent Variables 

All respondents were asked to report lifetime substance use, including which, if any, of the following classic psychedelics they had ever used: psilocybin, DMT, ayahuasca, LSD, mescaline, peyote and San Pedro. Those respondents who reported lifetime classic psychedelic use were also asked to retrospectively rate the degree of ego dissolution during their most intense experience using a classic psychedelic. 

Ego dissolution was measured by using the eight-item Ego-Dissolution Inventory (EDI) [44]. The items were rated on a scale from 0 to 100 (0 = No, not more than usually, 100 = yes, entirely or completely): “*I experienced a dissolution of my ‘self’ or ego*”, “*I felt at one with the universe*”, “*I felt a sense of union with others*”, “*I experienced a decrease in my sense of self- importance*”, “*I experienced a disintegration of my ‘self’ or ego*”, “*I felt far less absorbed by my own issues and concerns*”, “*I lost all sense of ego*”, “*All notion of self and identity dissolved away*”. Higher scores indicated more ego dissolution. Internal consistency was excellent (*α* = 91).

#### 2.2.3. Control Variables

As specified in the pre-registration, control variables included age in years, gender, educational attainment, lifetime use of cocaine, and alcohol-related risk-behavior (measured with Alcohol Use Disorders Identification Test—Concise) [51], which broadly corresponded with covariates used in prior research on classic psychedelics and attitudes [47]). Other variables relevant to human–animal relations were also included (diet, pet ownership, political affiliation and nature visits). 

### 2.3. Statistical Analyses

We used linear regression models to evaluate associations between classic psychedelic-related variables and human–animal relations. The covariates in all pre-registered models included: age in years (18–25, 26–34, 35–49, 50–64 or 65 or older); gender (male, female, transgender/non-binary); educational attainment (some high school or less, high school graduate or equivalent, some college/community college degree, Bachelor’s degree or higher); lifetime use of cocaine (yes, no); alcohol-related risk-behavior (continuous); diet (omnivore, pescatarian, vegetarian, vegan); pet ownership (yes, no); political affiliation (Republican, Democrat, Independent, other, none); and nature visits (every day, one to several times per week, one to several times per month, one to several times per year, never).

## 3. Results

Table 1 presents results from the regressions testing the associations between classic psychedelic use, degree of ego dissolution during the most intense experience using a classic psychedelic, and human–animal relations. As demonstrated in the table, lifetime classic psychedelic use was negatively associated with speciesism (*β* = −0.07, *p* = 0.002) and positively associated with animal solidarity (*β* = 0.04, *p* = 0.041), but it was not associated with desire to help animals (*β* = 0.01, *p* = 0.542). Notably, ego dissolution during the respondents’ most intense experience using a classic psychedelic was negatively associated with speciesism (*β* = −0.17, *p* < 0.001) and positively associated with both animal solidarity (*β* = 0.18, *p* < 0.001) and desire to help animals (*β* = 0.10, *p* = 0.007).

Exploratory analyses were conducted to investigate the associations between lifetime classic psychedelic use, ego dissolution and nature visits. As can be seen in Table 2, both lifetime classic psychedelic use and ego dissolution were positively associated with the average number of visits to nature areas in the past 12 months (*β* = 0.06, *p* = 0.007, and *β* = 0.13, *p* = 0.002, respectively). 

## 4. Discussion

This study investigated the associations between lifetime classic psychedelic use, ego dissolution during the respondents’ most intense experience using a classic psychedelic, and attitudes toward animals, in a sample representative of the US adult population, with regard to age, sex and ethnicity. The results showed that lifetime classic psychedelic use and ego dissolution were negatively associated with speciesism and positively associated with animal solidarity. Ego dissolution, but not lifetime classic psychedelic use, was positively associated with the desire to help animals. Furthermore, exploratory analyses showed that lifetime classic psychedelic use and ego dissolution were also associated with more frequent nature visits during the past 12 months. These findings suggest that classic psychedelic use may impact human–animal relations, which corresponds with previous studies linking classic psychedelic use to an increase in nature relatedness, with effects dependent on ego dissolution [43,44,46] (see also [48]). 

If there is indeed a causal relationship between classic psychedelic use and human–animal relations, it is possible that the mechanism underlying the effects could be related to ego dissolution and its impact on defense mechanisms that may arise when dealing with cognitive dissonance. While these defense mechanisms can decrease negative feelings and discomfort, they may also inhibit learning processes and prevent attitudes and/or behaviors from emerging as a result of those negative but possibly important experiences [52]. It is plausible that ego dissolution could contribute to a re-connection to animals and a dismantling of (perceived and constructed) human–animal differences. Such differences are often used to highlight positive and distinctive traits amongst humans, to decrease identification with animals, and to increase a sense of superiority over animals and exploitation of the natural world [28,29]. When socio-culturally constructed value systems are dismantled, there is the potential for animals (especially non-pet animals), to be re-individualized, and not solely viewed as measures for human gains and industrial practices. For instance, previous research has found that classic psychedelic users—irrespective of culture of origin—score lower on the valuing of financial prosperity [53]. When the perceived human–animal gap decreases, the win-lose situation often portrayed in acting in favor of animals can diminish and be replaced by a win-win situation and a framework of interconnection. 

There is increasing awareness about the role of extra-pharmacological factors in the effects following classic psychedelic experiences [54,55,56,57,58]. For example, being in natural (non-man-made) surroundings during a classic psychedelic experience may strengthen the effects on nature-relatedness [43,57]. As classic psychedelics appear to increase neuroplasticity [59,60], users may become more sensitive to exposure of stimuli during and some time after a classic psychedelic experience. There is a need to emphasize and further explore the importance of context for classic psychedelic experiences. In this case, this can include the role of animal-related settings (e.g., contact with or presence of animals), but also the impact of socio-cultural and economic factors in our immediate surroundings that reinforce disassociation from animals, prejudicial tendencies and defense mechanisms, to rationalize animal exploitation on a daily basis (e.g., industrial practices disconnecting the animal being from the final product, through language and packaging [24]). Thus, facilitating positive relations with animals before and after the use of classic psychedelics could potentially amplify the effects of classic psychedelic use on human–animal relations, and decrease the human–animal divide, with the potential thereby to create a long-lasting connection to, and solidarity with, animals, which could subsequently encourage more pro-environmental attitudes and behaviors.

There are several limitations in this study that need consideration. Firstly, as the study is based on cross-sectional data, the findings cannot establish a cause-and-effect relationship between classic psychedelic-related variables and human–animal relations. The self-reported data also raises questions about response biases, as well as if, and how, connection to animals might be translated into real-life action. Objective measures and longitudinal study designs are needed to complement the results in this study. Future studies could also include engagement-related outcomes, including pro-sociality and compassion [61,62]. Secondly, the regression models controlled for a range of potential confounders, but the associations may have been affected by other confounding variables that were not included in this study (e.g., disorders or medication that may induce similar sensations to that of ego dissolution [42]). Thirdly, whereas data was collected on other types of substance use (e.g., lifetime cocaine use), there was no data collected on the cultural context and setting in which the classic psychedelic experience took place. Fourthly, as data was only collected from US residents, the findings cannot be generalized to populations in other countries. Fifthly, the study did not include specific questions about types of animals. It is therefore unclear what kind of “animal connection” is being explored, what “connection” means, and how this can depend on the perceived moral and hierarchical status of different animals. There is a need to further understand a person’s sense of their place in the natural order because, interestingly, people can see themselves as part of nature, yet define nature as free of human presence and contact [9]. Due to the present construct of human–animal relations, connection to and solidarity with animals can still involve some sort of disassociation and categorization, and not always translate into action.

The interest in psychedelics as a nature-connecting agent for individuals is growing, and brings multiple challenges requiring attention, including: biomedical and cultural misappropriation, and historically rooted (colonial) assumptions regarding indigenous communities and classic psychedelic use [63,64]; the commodification of classic psychedelics and the risk of over-exploiting natural resources and disrupting traditional social systems [65,66]; and the risk of a continuous focus on individualism and consumption-based solutions [67]. Notably, classic psychedelics have been used for centuries as part of the traditional medicine of many cultures [64], and knowledge of, and worldviews with, interspecies connection already exist [67,68]. The questions that are brought up in this study reflect the ambition, within sustainability research, to explore approaches that include human worldviews, mindsets, relations, culture and individual behaviors as complementary to the dominant technical and external framing of the environmental discourse [69], which was also similarly called for in a recent recommendation from the IPCC [33].

## 5. Conclusions

While previous research has found associations between lifetime classic psychedelic use and nature-relatedness, this study explored the associations between lifetime classic psychedelic use, ego dissolution during respondents’ most intense experience using a classic psychedelic, and attitudes toward animals. Results suggest that classic psychedelic use and ego dissolution may facilitate a shift in human–animal relations toward less anthropocentric frameworks that acknowledge human–animal-environmental interconnectedness and enable more-than-human solidarity, which may subsequently lead to an increase in pro-environmental and sustainable attitudes and behaviors [35,36,70,71,72]. More research is needed to better understand if, for whom, and under what circumstances classic psychedelic use might alter human–animal relations.

## Figures and Tables

**Table 1 ijerph-19-08114-t001:** Lifetime classic psychedelic use, ego dissolution, and human–animal relations.

	Speciesism		Animal Solidarity		Desire to Help	
	*β*	*p*	*β*	*p*	*β*	*p*
Lifetime classic psychedelic use	−0.07	0.002	0.04	0.041	0.01	0.542
Ego dissolution	−0.17	<0.001	0.18	<0.001	0.10	0.007

*β* = standardized coefficients; *β* are adjusted for age in years, gender, educational attainment, lifetime use of cocaine, alcohol-related risk behavior (continuous), political affiliation, pet ownership, diet and nature visits.

**Table 2 ijerph-19-08114-t002:** Lifetime classic psychedelic use, ego dissolution and nature visits.

	Nature Visits	
	*β*	*p*
Lifetime classic psychedelic use	0.06	0.007
Ego dissolution	0.13	0.002

*β* = standardized coefficients; *β* are adjusted for age in years, gender, educational attainment, lifetime use of cocaine, alcohol-related risk behavior (continuous), political affiliation, pet ownership and diet.

## Data Availability

Data and syntax are available at figshare: https://doi.org/10.6084/m9.figshare.20199407.v1 [Data] and https://doi.org/10.6084/m9.figshare.20199596.v1 [Syntax].

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
