# Peer review of "Classic Psychedelics and Human–Animal Relations"

_ijerph, 2022, doi:10.3390/ijerph19138114_

Round 1

Reviewer 1 Report

From the early 1960s–1970s, psychedelic drug-assisted psychotherapy was researched in the United States as a treatment for cancer-related psychological and existential distress. In February 1964, nurse Kay Parley published the article "Supporting the patient on LSD Day" describing how the "psychedelic experience" led people to “scan their life”, providing also new perspectives on lifetime past events and experiencing a wide range of emotions often leading to self-healing and self-understanding, too. Building upon this old research, several recently published trials examining psilocybin - a psychedelic drug - to treat cancer-related psychological and existential distress, and rapid and sustained improvements have been demonstrated. Nowadays, the USA and also an increasing number of countries worldwide (Israel, Canada, Switzerland, etc) are experiencing a "psychedelic renaissance”: mounting public interest in psychedelic research is reflected in the 2020 Global Drug Survey which stated that nearly 6% of 110,000 respondents used psychedelics in the past year for self-treatment of mental health conditions, and 90% who had a supervised psychedelic experience in an uncontrolled setting indicated interest in taking psychedelics under a legally regulated and approved treatment system. These findings underscore the need for increased public education, training of qualified care providers, and harm reduction approaches as regulatory frameworks evolve. On this perspective, the article "Classic psychedelics and human-animal relations" offers the opportunity to further spread the knowledge - and also to reduce the stigma - about the use of psychedelic drugs.

The under-review manuscript appears to be adequately clear and quite consistent in its form and structure. The experimental design is appropriate to test the primary hypothesis, and results appear to be sufficiently reproducible. Conclusions are consistent with the evidence and the arguments presented.

Nevertheless, the accuracy of the manuscript appear to be partly affected by the underestimation of some relevant aspects:

- What is Ego Dissolution? since the concept of ego dissolution turns to be a core point through the presented article, a deeper explanation is needed: how is it associated with psychedelic drug use? how does it develop? are there any related somatic symptoms which can help people to subjectively and objectively recognise it? what are the condition - other than psychedelic use - that may cause ego dissolution? To mention a few, previous psychiatric conditions (panic-anxiety disorders, psychotic disorders, etc) or neurological disorders (temporal epilepsy) may sometimes cause or predispose a similar syndrome.

- Control variables. The above mentioned psychiatric conditions and neurological disorders might appear useful to be assessed, in order to reduce the confounding effect on the sample. Also, some medication - namely serotoninergic compounds such as SSRIs - may cause dissociative symptoms hardly to be differentiate from the ego dissolution.

Furthermore, as it has been clearly established in the article, the study does not provide any evidence of a cause-and-effect relationship. Anyway, deeper analysis on anxiety/depression spectrum symptoms, perceived wellness and quality of life - through a proper test assessment on the sample both pre- and post-psychedelic drug use - may provide wider information on this research direction, whereas a proven improvement on these conditions might perhaps enlarge the attention of scientific community and public opinion on the necessity of a sort of "psychedelic revolution".

Author Response

We thank Reviewer 1 for the valuable feedback we have received on our manuscript “Classic psychedelics and human-animal relations” for the International Journal of Environmental Research and Public Health. We have incorporated changes that are highlighted within the manuscript by using track changes. Here is a point-by-point response to the reviewer’s comments and concerns.

· Comment 1: What is Ego Dissolution? since the concept of ego dissolution turns to be a core point through the presented article, a deeper explanation is needed: how is it associated with psychedelic drug use? how does it develop? are there any related somatic symptoms which can help people to subjectively and objectively recognise it? what are the condition - other than psychedelic use - that may cause ego dissolution? To mention a few, previous psychiatric conditions (panic-anxiety disorders, psychotic disorders, etc) or neurological disorders (temporal epilepsy) may sometimes cause or predispose a similar syndrome.

Response: As suggested by comment 1, we have written additional sentences on ego dissolution in the revised article, which can be found on page 2 (last paragraph) and page 3 (paragraph one) in the introduction. Whilst we agree on the importance of increasingly understanding the meaning (/meanings) and manifestations of ego dissolution, we believe that the explanation in the revised text is sufficient for the reader’s ability to understand and contextualize the findings of our research without going beyond the scope of our article.

· Comment 2: Control variables. The above mentioned psychiatric conditions and neurological disorders might appear useful to be assessed, in order to reduce the confounding effect on the sample. Also, some medication - namely serotoninergic compounds such as SSRIs - may cause dissociative symptoms hardly to be differentiate from the ego dissolution.

Response: We agree and have addressed this comment by adding additional confounding effects in the Limitations section of the paper, page 6.

· Comment 3: Furthermore, as it has been clearly established in the article, the study does not provide any evidence of a cause-and-effect relationship. Anyway, deeper analysis on anxiety/depression spectrum symptoms, perceived wellness and quality of life - through a proper test assessment on the sample both pre- and post-psychedelic drug use - may provide wider information on this research direction, whereas a proven improvement on these conditions might perhaps enlarge the attention of scientific community and public opinion on the necessity of a sort of "psychedelic revolution".

Response: Thank you for this insightful comment regarding psychedelic research developments and future, where methodology and data are of essence in order to establish cause-and-effect relationships. We do believe we address the limitations of our study in sufficient detail in the discussion, including the implications of cross-sectional data and the need for future research that use objective measures and longitudinal study. 

Reviewer 2 Report

Review on the manuscript of Pöllänen E. et al.: “Classic psychedelics and human-animal relations”.

This manuscript explores the relationship between psychedelic use, the most intense experience of ego dissolution during the consumption of psychedelics, and attitudes towards animals. The authors found that the use of psychedelic compounds was negatively associated with speciesism and positively associated with animal solidarity. In addition, as reported by the individuals included in the study, the most intense experience of ego dissolution during the consumption of psychedelics was also negatively associated with speciesism, but positively associated with animal solidarity and wish to help animals.

The data shown in the manuscript seem to be clear and well explained. Some minor issues that arise are listed below for consideration of the authors.

1 - Is there any reason to perform the study in the US population? Does the incidence of psychedelics consumption is higher in the US population?

2 - Since the manuscript is relatively short, I would recommend the authors to include the supplementary table in the main text of the manuscript.

Author Response

We thank Reviewer 2 for useful comments that we have taken into account when revising our manuscript “Classic psychedelics and human-animal relations” for the International Journal of Environmental Research and Public Health. Changes made that reflect reviewer two’s feedback can be found highlighted by using track changes, and also described and addressed below in a point-by-point response.

· Comment 1: Is there any reason to perform the study in the US population? Does the incidence of psychedelics consumption is higher in the US population?

Response: The prevalence of lifetime classic psychedelic use is known in the United States, but limited data exists on populations in other countries. In addition, Prolific Academic only offers representative data for US and UK populations. For these reasons, we found it reasonable to conduct the study with a US sample. We have considered this clarifying comment and added an additional text in the Section Limitations on page 6 and page 7.

· Comment 2: Since the manuscript is relatively short, I would recommend the authors to include the supplementary table in the main text of the manuscript.

Response: In accordance with the second comment, the supplementary table can now be found in the main text of the manuscript as Table 2 on page 5 after Table 1. Additionally, corrections and additions have been made in the main text referring to Table 2, on the same page, in the paragraph above the two tables.